# Feasibility, SAR Distribution, and Clinical Outcome upon Reirradiation and Deep Hyperthermia Using the Hypercollar3D in Head and Neck Cancer Patients

**DOI:** 10.3390/cancers13236149

**Published:** 2021-12-06

**Authors:** Michiel Kroesen, Netteke van Holthe, Kemal Sumser, Dana Chitu, Rene Vernhout, Gerda Verduijn, Martine Franckena, Jose Hardillo, Gerard van Rhoon, Margarethus Paulides

**Affiliations:** 1Department of Radiation Oncology, Erasmus MC Cancer Institute, 3015GD Rotterdam, The Netherlands; tenvijver@xs4all.nl (N.v.H.); k.sumser@erasmusmc.nl (K.S.); r.vernhout@erasmusmc.nl (R.V.); g.verduijn@erasmusmc.nl (G.V.); m.franckena@erasmusmc.nl (M.F.); g.c.vanrhoon@erasmusmc.nl (G.v.R.); m.m.paulides@tue.nl (M.P.); 2Holland Proton Therapy Center, 2629JH Delft, The Netherlands; 3Department of Hematology, HOVON Data Center, Erasmus MC Cancer Institute, 3015GD Rotterdam, The Netherlands; d.chitu@erasmusmc.nl; 4Department of Otorhinolaryngology—Head and Neck Surgery, Erasmus MC Cancer Institute, 3015GD Rotterdam, The Netherlands; j.hardillo@erasmusmc.nl; 5Department of Electrical Engineering, Eindhoven University of Technology, 5612AZ Eindhoven, The Netherlands

**Keywords:** head and neck cancer, hyperthermia, reirradiation, treatment outcome

## Abstract

**Simple Summary:**

Following radiotherapy for head and neck cancer, patients are at risk for developing a recurrent or second tumor. Often reirradiation is required in these patients, which is hampered in dose by the previous irradiation. Besides chemotherapy, hyperthermia can potentially increase the effectivity of the radiotherapy. In this study we have used a new hyperthermia applicator in order to increase the effectivity of the radiotherapy in patients requiring reirradiation. We show that the added hyperthermia treatment is tolerated by patients and that we reach a higher hyperthermia dose to the tumor compared to the previous applicator. In addition, we show that the tumor control and survival as well as toxicity are similar compared to what has been reported in literature using chemotherapy as an additive to reirradiation in head and neck cancer patients.

**Abstract:**

(1) Background: Head and neck cancer (HNC) patients with recurrent or second primary (SP) tumors in previously irradiated areas represent a clinical challenge. Definitive or postoperative reirradiation with or without sensitizing therapy, like chemotherapy, should be considered. As an alternative to chemotherapy, hyperthermia has shown to be a potent sensitizer of radiotherapy in clinical studies in the primary treatment of HNC. At our institution, we developed the Hypercollar3D, as the successor to the Hypercollar, to enable improved application of hyperthermia for deeply located HNC. In this study, we report on the feasibility and clinical outcome of patients treated with the Hypercollar3D as an adjuvant to reirradiation in recurrent or SP HNC patients; (2) Methods: We retrospectively analyzed all patients with a recurrent or SP HNC treated with reirradiation combined with hyperthermia using the Hypercollar3D between 2014 and 2018. Data on patients, tumors, and treatments were collected. Follow-up data on disease specific outcomes as well as acute and late toxicity were collected. Data were analyzed using Kaplan Meier analyses; (3) Results: Twenty-two patients with recurrent or SP HNC were included. The average mean estimated applied cfSAR to the tumor volume for the last 17 patients was 80.5 W/kg. Therefore, the novel Hypercollar3D deposits 55% more energy at the target than our previous Hypercollar applicator. In patients treated with definitive thermoradiotherapy a complete response rate of 81.8% (9/11) was observed at 12 weeks following radiotherapy. Two-year local control (LC) and overall survival (OS) were 36.4% (95% CI 17.4–55.7%) and 54.6% (95% CI 32.1–72.4%), respectively. Patients with an interval longer than 24 months from their previous radiotherapy course had an LC of 66.7% (95% CI 37.5–84.6%), whereas patients with a time interval shorter than 24 months had an LC of 14.3% (95% CI 0.7–46.5%) at 18 months (*p* = 0.01). Cumulative grade 3 or higher toxicity was 39.2% (95% CI 16.0–61.9%); (4) Conclusions: Reirradiation combined with deep hyperthermia in HNC patients using the novel Hypercollar3D is feasible and deposits an average cfSAR of 80.5 W/kg in the tumor volume. The treatment results in high complete response rates at 12 weeks post-treatment. Local control and local toxicity rates were comparable to those reported for recurrent or SP HNC. To further optimize the hyperthermia treatment in the future, temperature feedback is warranted to apply heat at the maximum tolerable dose without toxicity. These data support further research in hyperthermia as an adjuvant to radiotherapy, both in the recurrent as well as in the primary treatment of HNC patients.

## 1. Introduction

Recurrent or second primary (SP) head and neck cancer (HNC) after radiotherapy occurs in 30–40% of patients [1,2,3,4,5]. Treatment of previously irradiated patients is a clinical challenge to date, especially when tumors are inoperable, as both the recurrent tumor as well as the renewed radiotherapy course carry substantial risks of morbidity and mortality [5]. Historically, reirradiation with or without sensitizing chemotherapy resulted in poor locoregional control rates [6,7]. In more recent literature, however, treatment with definitive reirradiation with or without chemotherapy showed a local control (LC) rate of 42.7% and an overall survival (OS) of 35.5% at 2 years [5]. Therefore, it seems that in the current era of radiotherapy techniques, reirradiation is becoming a more realistic treatment option for recurrent HNC patients, although careful selection seems warranted [8].

In reirradiation of HNC, chemotherapy is commonly used to sensitize radiotherapy. However, chemotherapy can result in increased side effects from the radiotherapy and carries potential systemic side effects, limiting its use in patients with comorbid disease [7]. Clinical hyperthermia represents an alternative to chemotherapy as a sensitizer of radiotherapy. In primary HNC, elevation of target temperatures to 40–44 °C results in around 20 percent increase in LC [9]. In recurrent HNC, however, the effect of adjuvant hyperthermia to radiotherapy has not been explored thoroughly [2,10,11].

In primary HNC patients, hyperthermia is mostly delivered using capacitive or intraluminal heating devices [12]. These devices can only adequately heat superficial tissues, and treatments in these studies were mostly applied without real-time monitoring. To be able to heat deep-seated tumors and to better steer the energy deposition, we previously developed a medical hyperthermia device incorporating 12 antennas, named the Hypercollar, that can focus microwaves to the target volume [13]. In addition, we developed and validated 3D simulation technology to optimize settings in pretreatment planning and for real-time simulation guided treatment and control [14,15]. We have previously reported on the safety and feasibility of treatment with the Hypercollar [10]. Learning from this experience, we further developed the Hypercollar for improved applicability, patient comfort, and energy steering [12]. This next-generation device, named the Hypercollar3D, has 20 antennas and an improved water bolus fitting for improved heating of the oropharyngeal and nasopharyngeal areas [16]. The better fit also creates a better match between simulation and treatment to improve the simulation-guided treatment.

Since the clinical introduction of the Hypercollar3D in 2014, we have treated 22 patients with recurrent or SP HNC receiving reirradiation with curative intent. The goal of this study was to evaluate the feasibility, acute and late toxicity as well as the clinical outcome in recurrent or SP HNC patients following thermoradiotherapy using the Hypercollar3D.

## 2. Materials and Methods

### 2.1. Patient Population

The research protocol for this retrospective study was reviewed by the medical ethics committee of Erasmus MC Cancer Institute, Rotterdam (MEC-2018-1453) and was classified as not falling within the definition and scope of the WMO (Medical Research Involving Human Subjects Act). Patients included were treated at our institute between 2014 and 2018 for a recurrent or SP HNC with reirradiation combined with deep hyperthermia using the Hypercollar3D with curative intent. Exclusion criteria for deep hyperthermia were systemic temperatures of >39 °C, claustrophobia, tumor caudal to a tracheostomy (this prevents penetration of the microwaves to the tumor), anatomical boundaries of the shoulders prohibiting positioning of the applicator, and the presence of a pacemaker.

### 2.2. Radiotherapy Treatment

Radiotherapy technique, radiation field, dose, and fractionation were left at the discretion of the treating radiation oncologist and are listed in Table 1. In brief, radiation fields included at least the primary tumor site with or without elective neck irradiation. Radiotherapy techniques used were stereotactic radiotherapy using the Cyberknife (Accuracy Inc., Sunnyvale, CA, USA) in 7 patients and external beam radiotherapy (IMRT or VMAT) in 15 patients. Fractionation schemes are listed in Table 1.

### 2.3. Hyperthermia Treatment

Hyperthermia (HT) was delivered following the radiotherapy fraction and was delivered weekly. The target volume for hyperthermia was the gross tumor volume (GTV) with a margin to account for planning and positioning inaccuracies. In the postsurgical situation, usually the clinical target volume (CTV) for radiation was the target volume for hyperthermia. If this was too large for adequate heating, a high-risk zone was identified, and the truly elective areas were not primarily heated. For each patient, a 3D patient model was generated by applying automatic segmentation of the planning computed tomography (CT) scan from the radiotherapy treatment [17]. Next, the patient model was imported into SEMCAD-X (Zurich MedTech, Zurich, Switzerland) to calculate the electromagnetic field per antenna. The resultant electric field distributions were imported into in-house developed software VEDO for optimizing the specific absorption rate (SAR) distribution by maximizing the target hotspot quotient (THQ) [15]. THQ is expressed using a total hotspot volume of 1% of the total volume (THQ_1%) [15].

Treatment was started using the preoptimized settings, and total power was gradually increased until the target temperature, a patient indicated hotspot, or a SAR constraint in the masseter region was reached. [10]. As in the earlier protocol, placement of invasive catheters inside the tumor was mandatory in case of a low predicted treatment quality, being 25% iso-SAR coverage (TC25) smaller than 75%, and optional for a TC25 above 75%. In the latter case, placement was often deemed too risky or too troublesome for patients; therefore, no temperatures could be measured. The protocol also included optional measurements of normal tissue temperatures in case distinct hotspots were to be expected based on the predicted SAR distribution.

In all cases, treatment was monitored in real-time using the applied cubic filtered SAR (cf-SAR) estimations [12]. Hereto, the real-time measured power and phase of the signals applied to the antennas were extrapolated into a real-time estimated applied SAR using the pre-calculated electric fields per antenna. Re-optimization of SAR distribution during the treatment was conducted if the patient had discomfort due to hotspots, which was discriminated from other sources of discomfort by briefly turning off total power. The duration of each hyperthermia treatment was 75 min, and heating up to 43 °C in the target or up to the patient’s tolerance was applied, aimed at achieving 40–44 °C in the target region for 60 min.

### 2.4. Collection of Patient and Follow-Up Data

Patient, tumor, and treatment details were extracted from the patients’ files. Specific radiotherapy and HT treatment characteristics were extracted from treatment planning and other recording systems. Local recurrence, distant recurrence, survival status, date and cause of death, as well as acute and late toxicity data were extracted and/or retrieved from patient records, referring hospitals, general practitioners, and the civil registry. Toxicity was scored according to CTCAE v4 at baseline, end of radiotherapy treatment, and 3–4 and 12 months post-treatment. Grade 1 toxicities were not included in the analyses because they were considered unreliable due to the retrospective data collection. After evaluation of the first five patients, we decided to introduce measurements of the range of motion (ROM) of the jaw before and after each treatment, as the CTCAE v4 scale is very robust for measuring trismus.

### 2.5. Hyperthermia Treatment Parameters

Hyperthermia treatment characteristics were collected. The number of hyperthermia treatments, hyperthermia treatment duration, mean applied power, mean estimated applied cf-SAR in tumor, and HTP planning parameters (THQ, TC25, TC50, and TC75) were extracted from the HT treatment files. The hyperthermia treatment session was marked as prematurely aborted if the total duration of the treatment was less than 70 min, since this indicates a problem reported by the patient. The effective treatment time per session was defined as the input power values higher than 1W during the treatment duration. The reported maximum estimated applied cf-SAR in tissues at risk were calculated retrospectively.

### 2.6. Statistical Analysis

LC and OS were calculated from the start date of radiotherapy until the event. LC was noted as ‘failed’ when a physician diagnosed a local recurrence either clinically or with imaging (CT/MRI). Patients were censored for LC after the last visit of any physician specifically examining for recurrent disease or death. For OS, patients were censored after the day the civil registry was consulted. LC and OS were analyzed using the Kaplan–Meier method and statistical differences between groups were determined using the log-rank test. A *p*-value of ≤0.05 was considered statistically significant. All analyses were performed using Stata 15.1 (StataCorp. 2017. Stata Statistical Software: Release 15, StataCorp LLC, College Station, TX, USA).

## 3. Results

### 3.1. General Characteristics of Patients, Treatment, and Follow-Up

Patient and radiotherapy characteristics are listed in Table 1. Median follow-up for local recurrence was 17.5 months (IQR 6.0–31.0 months) and for overall survival 24.5 months (IQR 11.0–48.0 months). Thirteen out of twenty-two patients had a local recurrence, fourteen out of twenty-two had any recurrence, and fourteen out of twenty-two died during follow up.

### 3.2. Hyperthermia Treatment Feasibility

Comparing the clinical performance of the Hypercollar3D with our earlier Hypercollar applicator, we note that the comfort of the treatment as experienced by the patient is comparable, i.e., for the Hypercollar3D applicator in 90% of the treatments the treatment duration was at least 70 min of the intended 75 min, compared to 87% for the earlier Hypercollar design.

Table 2 shows that in total 134.9 W of mean power (range = 49.9–353.0 W) was applied to achieve a mean predicted cfSAR of 104.2 W/kg (range = 36.5–314.8) in the target regions that had a mean volume of 40.8 cc (range = 2.8–233.9 cc). Treatment planning predicted a mean TC25 of 88%, TC50 of 55%, TC75 of 12% and THQ was on average 1.28. In 5/22 patients, temperatures were measured during the first hyperthermia session in the target (4.6%). Measured minimum (“starting”) temperatures ranged between 34.3–36.6 °C. This may be an indication that measurements were in general taken very superficially. Measured median temperature (T50) was 39.6 °C (37.2–41.9 °C) but a mean increase of 3.9 °C (1.3–6.2 °C) was achieved.

Because of the incidence of trismus (see toxicity below), after five patients the clinical protocol previously described by Verduijn et al. was adjusted by: (1) decreasing the power increments from 20 W to 10 W per 5 min of treatment time to allow for thermoregulative physiological adjustment, and (2) by applying an absolute SAR threshold of 175 W/kg in the masseter region. Following the adapted instructions, the mean applied power decreased from 278.8 W (first five patients) to 92.5 W (17 patients treated thereafter), and consequently, the mean estimated applied cfSAR in the tumor decreased from 185.0 W/kg to 80.5 W/kg.

In 6/22 patients one or more hyperthermia sessions were prematurely aborted. In four patients this was due to pain from the hyperthermia treatment, and in one of those patients it was due to an increase in pre-existing neuropathic pain. In addition, the last hyperthermia session was not applied in one patient due to tumor progression and for another patient it was due to claustrophobia and fear of technical problems. In 88% of the treatments, the treatment duration was at least 70 min of the intended 75 min. For all patients and treatments, the mean hyperthermia treatment time delivered, reaching 71.5 min or 95.3% of the intended time (75 min).

### 3.3. Oncological Outcome and Relation with Interval with Previous Radiotherapy

Thirteen patients were treated with definitive thermoradiotherapy, whereas, nine patients were treated with postoperative thermoradiotherapy (Table 1). In patients treated with definitive thermoradiotherapy a complete response rate of 81.8% (9/11; two not recorded) was observed (Table 1). Overall 2-year LC and OS were 36.4% (95% CI 17.4–55.7%) and 54.6% (95% CI 32.1–72.4%), respectively (Figure 1). Comparing postoperative versus definitive thermoradiotherapy, no significant difference was observed in 2-year LC; 33.3% (95% CI 7.8–62.3%) versus 38.5% (95% CI 14.1–62.8%), respectively (*p* = 0.97) (Figure 2).

The time interval between the first and the subsequent radiotherapy courses was previously reported to predict clinical outcome [5]. The median time from previous radiotherapy in our cohort was 51.5 months (IQR 17.5–122.0 months). Similar to observation by others, an interval of >24 months was significantly associated with higher LC; 18-month LC was 14.3% (95% CI 0.7–46.5%) for an interval of <24 months versus 66.7% (95% CI 37.5–84.6%) for an interval of >24 months (*p* = 0.01) (Figure 3).

### 3.4. Toxicity

The overall incidence of late grade 3 or higher toxicity at 2-years using Kaplan-Meier analysis was 39.2% (95% CI 16.0–61.9%), and the combined incidence of late grade 3 or higher toxicity or local recurrence at 2-years was 81.0% (95% CI 69.2–92.8%) (Figure 4). A detailed overview of type and grade of the toxicity at baseline, shortly after the radiotherapy course, 3–4 and 12 months is provided in Table 3. Of note, no grade IV or V toxicities were observed.

Notably, after evaluation of the first five patients treated with the HyperCollar3D, we saw that three patients had developed clinically relevant trismus; one patient developed grade I and two patients developed grade II trismus relatively early on in the course of treatment as an unexpected side effect. One of them also experienced grade III vertigo after the first treatment and a grade II vertigo after consecutive treatments. Two patients also developed a grade II edema of the neck. We adjusted the clinical protocol after our first evaluation (see feasibility) as explained earlier. After the introduction of the adapted treatment protocol, the occurrence of treatment-induced trismus at the end of the treatment decreased from 3/5 to 5/17. In the group of patients treated following the adapted protocol, the incidence of newly induced trismus at 3–4 months decreased further to 1/17 patients. For the other four patients with trismus at 3–4 months post treatment, two were in the first five patients having a high SAR value at the masseter and two already had trismus at start of the treatment and were not altered in grade by the treatment (Table 3).

## 4. Discussion

In this retrospective cohort study, we found that application of deep hyperthermia using the Hypercollar3D is feasible, and the oncological outcome is similar to other patient series of reirradiation with or without chemotherapy in HNC patients [5]. This study warrants further clinical studies using thermoradiotherapy in recurrent as well as in primary HNC patients.

In a meta-analysis in 2016 from Datta et al., five randomized trials and one nonrandomized trial, comparing radiotherapy versus radiotherapy plus hyperthermia in non-surgical patients, were analyzed [9]. This meta-analysis showed an overall improvement of the complete remission rate going from 39.6% (range 31.3–46.9%) with radiotherapy alone to 62.5% (range 33.9–83.3%) with radiotherapy plus hyperthermia [9]. Notably, the radiotherapy and hyperthermia treatments used in these studies were mostly performed using less advanced techniques compared to the current hyperthermia technique and/or applicator [18,19]. The CR rate of 81.8% for reirradiation plus hyperthermia observed in nine patients with definitive thermoradiotherapy is in the same range of the literature values, although patient selection and radiotherapy dosage may differ. This indicates the potency of hyperthermia to sensitize radiotherapy, also in the setting of reirradiation. This potency is well-studied in breast cancer, showing again higher complete response rates when hyperthermia is added to radiotherapy [20]. There are few studies reporting on the addition of hyperthermia to radiotherapy in recurrent or SP HNC [11,21]. Recently, we have published the clinical outcome using our previous head and neck applicator, the Hypercollar, in 27 HNC patients, including 18 recurrent or SP HNC patients [10]. In these latter patients, we observed a 2-year LC and OS of 36% and 33%, retrospectively. In our current cohort, we found similar 2-year LC and OS rates of 37% and 53%, respectively.

Comparing the outcome of our current cohort to recurrent HNC patients treated with chemoradiotherapy, several reported studies should be considered. In a randomized trial, albeit in the postoperative setting, full dose reirradiation with chemotherapy resulted in a 2-year LC of 60% [22]. In a recent cohort using IMRT, 2-year LC following reirradiation was reported as high as 64% [8]. More recently, a large, ‘real-world practice’ study by Ward et al. analyzed 412 recurrent or SP HNC patients [5]. Local regional control rate at 2 years was 42.7% for definitive reirradiation with or without chemotherapy [5]. We observed a local control rate of 37.5% at 2 years. In agreement with the data of Ward et al., we also observed a significantly worse local control when reirradiation was applied within 24 months from the previous radiotherapy course. Our data and the data from Ward et al. collectively support the careful selection of patients on the basis of the time interval with the previous radiotherapy course in order to prevent treatment related toxicity in patients having a dismal prognosis.

Cumulative Grade 3 or higher late toxicity was 33.5% in the study by Ward et al. and 39.2% in our cohort. Most of the important late toxicities were local, including xerostomia, dysphagia, osteoradionecrosis and trismus (Table 3). There were no grade IV or V toxicities and, although not measured, no systemic toxicities are to be expected from the hyperthermia treatment. Thus, the local toxicity following reirradiation is substantial but does not seem to be higher compared to reirradiation with or without chemotherapy [5]. In addition, hyperthermia is not expected to induce the potential systemic side effects of chemotherapy, like gastrointestinal, renal, neurological, and hematological side effects [23,24].

The average mean estimated applied cfSAR to the tumor volume for the Hypercollar3D was 80.5 W/kg for the last 17 patients treated with the adapted protocol, which is 55% higher than the 52 W/kg achieved with our earlier Hypercollar applicator. The gain in the improved cfSAR value achieved at the tumor volume can be explained by the fact that for the Hypercollar3D treatment, the VEDO software selects 12 antennas that make the largest contributions to the cfSAR at the tumor volume from the 20 available antennas. Further, the improved design of the water bolus, i.e., improved shape retention, results in a more efficient transfer of the EM energy.

With respect to hyperthermia related toxicity, the incidence of acute trismus grade II in three of the first five patients treated with the Hypercollar3D prompted us to an early evaluation of our treatment protocol. The adapted treatment protocol effectively reduced incidence of grade II trismus, but also resulted in a lower energy deposition in the target region from 185 W/kg to 81 W/kg. As invasive thermometry was not mandatory in our protocol, the effect of our protective measure for the normal tissues on the thermal dose to target could not be determined. We were also not able to unequivocally determine the safety of the applied reduction in energy deposition. Nevertheless, the fact that the late toxicity and oncological outcomes in our cohort were comparable with those reported in literature was reassuring. Further research is needed to confirm the safety and validity of the SAR thresholds used. For example, by introducing more extensive online temperature modeling whereby during treatment, temperatures are measured at several noninvasive reference locations. These measurements could be considered as a reference to calibrate predicted tissue temperatures for tumors and organs at risk. This could result in further treatment optimization and potentially prevent under treatment of the target area.

The feasibility to apply deep hyperthermia in HNC patients, as shown by our study, has potential implications for future clinical research. Besides recurrent or SP HNC patients, there are two important HNC patient groups that may especially benefit from thermoradiotherapy as opposed to chemoradiotherapy or radiotherapy alone [22]. The first group is HNC patients over 70 years old, in whom the additive effect of chemotherapy has not been demonstrated [23,24,25,26]. The second group is patients with a human papilloma virus (HPV) associated HNC, having a more favorable prognosis compared to HPV negative HNC patients [27,28]. Efforts are ongoing to de-escalate current treatment with radiotherapy by decreasing the dose. For example, in a phase 2 trial, HPV positive HNC patients were treated with induction chemotherapy. Complete or partial responders received only 54 Gy, resulting in a favorable progression free survival of 92% at 2 years [29]. The induction chemotherapy, however, induced grade III leucopenia and neutropenia in 39% and 11% of patients, retrospectively. Also in this report, the toxicity of deep hyperthermia of the head and neck is reported to be generally mild, and the radiotherapy toxicity was not enhanced in randomized trials [9,30]. One can hypothesize that adding hyperthermia to a reduced radiotherapy dose can result in a similar long term clinical outcome as the high dose radiotherapy but with less late toxicity from the radiotherapy.

Drawbacks of the current study are the retrospective nature with the potential for confounding data misinterpretation and missing data. In addition, the relatively small sample size, inhomogeneous patient group, and variability in radiotherapy and hyperthermia treatment and doses. Toxicity was not recorded in a standardized manner at our institute during follow-up. In our data collection process, however, toxicity was scored retrospectively by an independent data management team according to the CTCAE criteria and, when in doubt, the treating physician and an experienced head and neck surgeon were consulted (JAUH) to provide a final grading score. The optimal sequence and timing of the hyperthermia is still under debate. Similar to cervix carcinoma patients, we applied the hyperthermia after the radiotherapy fraction within four hours. In cervix carcinoma patients, this results in a clear thermal dose effect of the hyperthermia while the timing does not affect outcome, as long as the treatment is applied within four hours after radiotherapy [31,32]. Whether this is the optimal timing and whether these results can be extrapolated to HNC patients, is still a subject for future studies.

## 5. Conclusions

Deep hyperthermia using the Hypercollar3D in combination with reirradiation of recurrent and SP HNC is feasible in terms of patient tolerance and SAR deposition in the target area. The introduction of the Hypercollar3D resulted in an improved focused delivery with an increase in the SAR delivered to the tumor by 52 W/kg compared to the previous system. Good thermometry at target and normal tissues is needed to exploit this feature in future trials. The oncologic outcome as well as toxicity in our cohort were comparable to (chemo) radiotherapy in similar clinical settings. The data from our study are encouraging, although the relationship among SAR, tumor response, and toxicity requires further clinical validation. Our data warrant further prospective studies of deep hyperthermia using the Hypercollar3D. In the case of reirradiation, we are planning for a prospective registration study including, not only hyperthermia as an adjuvant, but also chemotherapy or reirradiation alone. In primary HNC we are aiming for a feasibility trial in patients who are not eligible for adjuvant chemotherapy. In this patient group we also aim to be able to have more invasive thermometry due to less pre-existent morbidity. When hyperthermia proves to be feasible in these patients and we have gained better understanding of the relation between SAR, temperature, and toxicity, a larger phase II clinical trial in primary HNC patients can be envisioned.

## Figures and Tables

**Figure 1 cancers-13-06149-f001:**
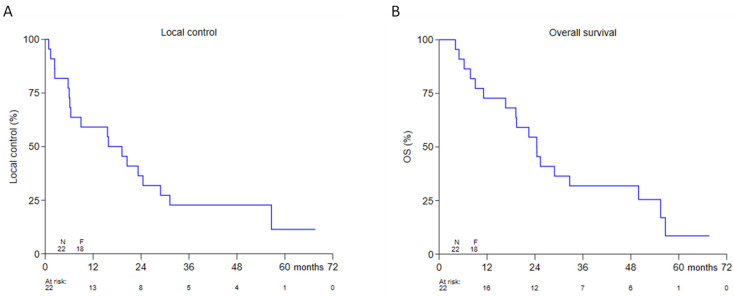
Kaplan-Meier analysis for LC (**A**) and OS (**B**) in 22 patients with a recurrent or second primary head and neck cancer.

**Figure 2 cancers-13-06149-f002:**
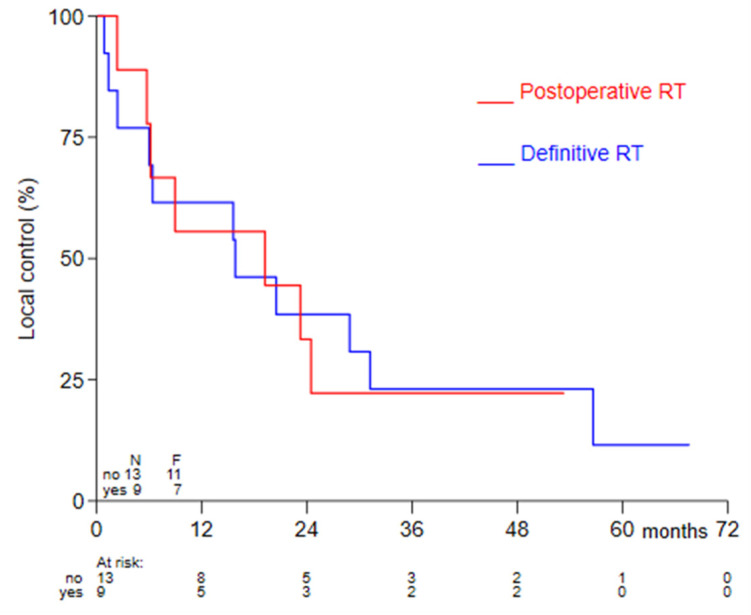
Kaplan-Meier curves for LC were compared for definitive versus postoperative treatment using the log-rank test.

**Figure 3 cancers-13-06149-f003:**
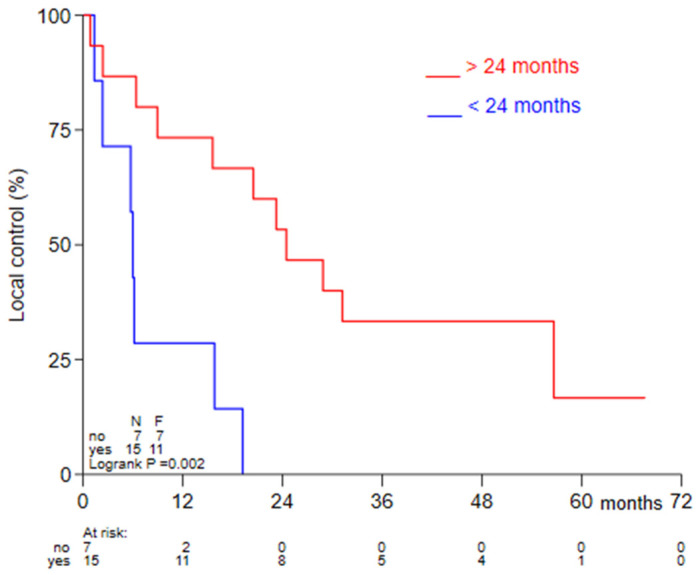
Kaplan-Meier curves for LC were compared for long (>24 months) and short (<24 months) time intervals between radiotherapy courses using the log-rank test.

**Figure 4 cancers-13-06149-f004:**
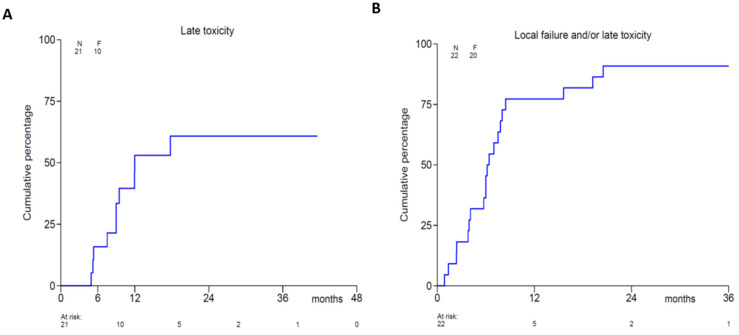
Kaplan-Meier analysis of late (>4 months) grade 3 or higher toxicity (**A**) and grade 3 or higher toxicity and/or local recurrence (**B**).

**Table 1 cancers-13-06149-t001:** General characteristics of patient tumor and treatments.

Characteristic	Categories	Value
Patient/tumor characteristics		
Age (years)	Years (median)	67.0 (IQR 59.5–71.5)
Sex	Male	16 (73.0%)
	Female	6 (27.0%)
Prior surgery (primary tumor)	Yes	13 (59.0%)
	No	9 (41.0%)
Prior systemic therapy (primary tumor)	Yes	7 (32.0%)
	No	15 (68.0%)
Recurrent/SP HNC	Recurrent tumor	14 (64.0%)
	Second primary tumor	8 (36.0%)
Tumor site (recurrent or SP tumor)	Nasopharynx	2 (9.0%)
	Oropharynx	12 (55.0%)
	Oral cavity	2 (9.0%)
	Salivary gland	2 (9.0%)
	Hypopharynx	1 (5.0%)
	Larynx	3 (14.0%)
Histology (recurrent or SP tumor)	Squamous cell carcinoma	19 (86.0%)
	Other	3 (14.0%)
Tumor stage (recurrent or SP tumor)	T0	8 (36.0%)
	T1	1 (5.0%)
	T2	6 (27.0%)
	T3	2 (9.0%)
	T4	4 (18.0%)
	Unknown	1 (5.0%)
Nodal stage (recurrent or SP tumor)	N0	9 (41.0%)
	N1	3 (14.0 %)
	N2	8 (36.0%)
	N3	1 (5.0%)
	Unknown	1 (5.0%)
Postoperative/definitivereirradiation + hyperthermia	Postoperative	9 (41.0%)
	Definitive	13 (59.0%)
Fractionation radiotherapy	6 × 5.5 Gy	7 (31.8%)
	10 × 2.0 Gy	1 (4.5%)
	25 × 2.0 Gy	2 (9.0%)
	28 × 1.8 Gy	1 (4.5%)
	30 × 2.0 Gy	9 (40.9%)
	33 × 1.8 Gy	2 (9.0%)
Technique radiotherapy	IMRT	10 (44.5%)
	VMAT	5 (22.7%)
	Cyberknife	7 (31.8%)
Radiation field	Tumor	10 (45.5%)
	Neck	7 (31.8%)
	Both	5 (22,7%)
Time from previous radiotherapy treatment	Months (median)	51.5 (IQR 17.5–122.0)
Number of planned hyperthermia treatments	Number of treatments per patient	Number of patients
	3	7 (31.8%)
	4	1 (4.5%)
	5	2 (9.1%)
	6	11 (50%)
	7	1 (4.5%)
Total number of all treatments	108	22 (100%)
Complete clinical response 12 weeks post-treatment for definitive radiotherapy	Yes	9 (81.8%)
	No	2 (18.2%)

**Table 2 cancers-13-06149-t002:** Hyperthermia treatment parameters.

Characteristic	Categories	Value
Hyperthermia treatment characteristics		
HT treatments	*n*	107
HTV volume	milliliters	40.8 mL (2.8–108.9)
Treatment planning		
TC25	%	90 (44–99)
TC50	%	58 (5–80)
TC75	%	12 (0–48)
THQ_1%	-	1.28 (0.38–3.83)
Mean applied power *	Watts	All 1–22: 134.9 (49.9–353.0)
		Pat 1–5: 278.8 (179.3–353.0)
		Pat 6–22: 92.5 (49.9–123.1)
Mean estimated cf-SAR tumor * (applied power * predicted cf-SAR * efficiency)	W/kg	All 1–22: 104.2 (36.5–314.8)
		Pat 1–5: 185.0 (69.4–314.8)
		Pat 6–22: 80.5 (36.5–145.1)
Target temperature	*n* (%)	5 (4.6)
Patient reference		A B C D E
Maximum	°C	38.3, 43.9, 40.9, 42.2, 38.0
Median	°C	37.8, 40.8, 40.5, 41.9, 37.2
Minimum	°C	36.5, 36.6, 35.8, 35.7, 34.3
Maximum normal tissue temperature	*n* (%)	56 (52.3)
Median	°C	40.1 (35.0–42.8)

* After the first five patients, a protocol adaptation lowering the energy in the masseter muscles was introduced.

**Table 3 cancers-13-06149-t003:** Description of toxicity at baseline, end of radiotherapy treatment, 3–4 months and 12 months post-treatment.

Toxicity		Baseline *N* = 22	End RT *N* = 18	3–4 Months Post-Treatment *N* = 18	12 Months Sost-Treatment *N* = 10
	Grade	Number (%)	Number (%)	Number (%)	Number (%)
Xerostomia	2	0	1 (6%)	1 (6%)	0
	3	0	1 (6%)	1 (6%)	1 (10%)
Altered taste	2	0	2 (11%)	1 (6%)	0
	3	0	0	0	0
Dysphagia	2	6 (27%)	6 (33%)	5 (28%)	5 (50%)
	3	3 (14%)	5 (28%)	3 (17%)	2 (20%)
Edema face	2	0	2 (11%)	0	1 (10%)
	3	0	0	0	0
Erythema skin	2	0	0	0	0
	3	0	0	0	0
Ulcus skin	2	0	1 (6%)	2 (11%)	2 (20%)
	3	0	0	0	1 (10%)
Trimus	2	2 (9%)	5 (28%)	4 (22%)	3 (30%)
	3			1 (6%)	1 (10%)
Osteoradionecrosis	Yes	0	0	3 (17%)	1 (10%)
	No	22 (100%)	18 (100%)	15 (83%)	9 (90%)
Burn wound	Yes	0	0	0	0
	No	22 (100%)	18 (100%)	18 (100%)	10 (100%)
Vertigo	Yes	0	0	0	0
	No	22 (100%)	18 (100%)	18 (100%)	10 (100%)
Tube feeding	Yes	2 (9%)	6 (33%)	3 (17%)	2 (20%)
	No	20 (91%)	12 (67%)	15 (83%)	8 (80%)
Opioid use	Yes	4 (18%)	9 (50%)	6 (33%)	1 (10%)
	No	18 (82%)	9 (50%)	12 (66%)	9 (90%)
Other grade 3 or higher toxicity	Yes	1 (5%)	1 (6%)	1 (6%)	1 (10%)
	No	21 (95%)	17 (94%)	17 (94%)	9 (90%)
Tracheostoma	Yes	4 (18%)	4 (22%)	1 (6%)	1 (10%)
	No	18 (82%)	13 (72%)	17 (94%)	9 (90%)

## Data Availability

The data presented in this study are available on request from the corresponding author. The data will be kept for at least 15 years after this publication at a secure location at Erasmus University Hospital in Rotterdam.

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
