# Peer review of "Feasibility, SAR Distribution, and Clinical Outcome upon Reirradiation and Deep Hyperthermia Using the Hypercollar3D in Head and Neck Cancer Patients"

_cancers, 2021, doi:10.3390/cancers13236149_

Round 1

Reviewer 1 Report

Kroesen et al. present a study performed between 2014 and 2018 to evaluate the feasibility and the clinical outcome of patients with recurrent or second primary  tumor treated with hyperthermia and radiotherapy. In this retrospective study, 22 patients were included and they concluded to an increased deposition of the energy of 55% with hypercollar 3D device compared with the previous system as well as a LC higher when the interval with the previous radiotherapy was longer than 24 months.  Hyperthermia combined with irradiation improve the curative potential of radiotherapy. Recently, technological advances in hyperthermia delivery and monitoring have been developed and are critical for implementing HT + RT in clinical practice. Many cancers, mainly superficial, are eligible to HT treatment and now deep-tissue tumors such as head and neck cancers. Combination of HT and Radiochemotherapy present interests in order to induce radiosensitizing responses. However, the treatment success depends on highly variable and flexible parameters such as treatment schedule and appropriate choice of radiation and thermal dose (defined by temperature and exposure time). In this study, these settings are very different between patients and not always detailed. Furthermore, a comparison with the previous device is made but with different settings and parameters of recruitment for patients. Patients are at different stages, different protocols of treatment which makes difficult to compare each groups. Furthermore, it seems that some mistakes are present in the tables.

Why the complete clinical response 12 weeks  post-treatment in postoperative radiohyperthermia is not developed in table 1 and elsewhere in the manuscript?

Twenty-two patients were included in this protocol. However, it looks in table 1 that 24 received a treatments based on hyperthermia. Please clarify. What are the 2 patients non recorded?

According table 1, they are 103 treatments applied + 2 non recorded. However, in table 2, authors talk about 107 HT treatments, which is not coherent with table 1 data.

In the general characteristics of patients, authors talk about 13/22 patients with local recurrence but 14/22 with any recurrence and 14/22 died. If there is 22 patients in total, how is it possible? Please look at these numbers and clarify.

For 5 patients among 22, at least one sessions was prematurely aborted. The number of session aborted for each patient has to be mentioned even if 96% of the intended time was applied.

Why OS are not given for the 2 groups of postoperative and definitive thermoradiotherapy?

Concerning Kaplan-Meier analysis for LC and OS in Figure 1, a control group wihout hyperthermia must be added to compare the benefit of HT + RT for HNC patients.

Figures 4 are very small and difficult to read.

Please correct the term “morfine use”.

A general comment concerns the title of the article wich is about feasibility, SAR distribution and clinical outcome using the hypercollar 3D device in HNC patients. However, the title does not really fit with the content since it is more about distribution, toxicities and outcomes. Please consider to modify it.

Reviewer 2 Report

Kroesen and co-workers presented a work that describe the use of Hypercollar3D for the treatment of recurrent or second primary head and neck cancer. Hypercollar 3D is an updated version of Hypercollar and the authors follow patients from 2014 to 2018. The work is interesting but is not clear how the new version of hypercollar improved the patients survival respect to the previous version. I suggest the authors to add a figure and/or table with a comparison between the two systems to better underline why the new version is better than the older one. My main concern regard the measure of the temeprature. How do the authors make sure that the correct temperature is reached during the treatment if they have not had the opportunity to measure it?

Minors:

Figures should be in high quality because very often it is hard to read labels.

Figure 4 is too small

Reviewer 3 Report

The authors do not overstate their conclusions and correctly note their limitations. 

The paper is well written.

  1. The main question addressed by the present research was one of feasibility of utilizing the Hypercollar3D for re-irradiation with hyperthermia which is why a control group is technically not necessary.
  2. There is some novelty in addressing the use of the Hypercollar3D compared to the Hypercollar
  3. The addition is also in use of this in recurrent HNC patients as this has been addressed more readily ni thoracic and breast cancer patients 
  4. The authors could have done a completely different study but that isn't really feasible; this struck me as more of a pilot study to test feasibility with aims of performing a larger study at a later date. If anything the authors should make clear any intentions they have for performing such a study. They are clear in their limitations of having a small sample size and having a mixed cohort of HNC types; in future the authors should either produce a matched cohort or limit to one cancer site (oropharyngeal, laryngeal, etc.). Yes, numbers are going to be a limitation in HNC studies, but numbers do not determine the science.
  5. Yes, the conclusions are appropriate and address the main questions posed
  6. Yes, the references are appropriate for a feasibility study
  7. No comments on the tables and figures

Reviewer 4 Report

Thank you for giving me the opportunity to review this exciting manuscript.

It is very well written and covers a medical need.

The data presented are impressive: In an intensively pre-treated group of patients with (mainly lymphogenically metastasised) recurrences of HNSCC, a cCR of 80% after 12 weeks is achieved by combining re-irradiation with hyperthermia. Such an effectiveness is not conceivable with the combination of radiation with chemotherapy.

I have a small request:

Could the authors please revise Figure 4 (toxicity) so that the Kaplan-Meyer curves are displayed larger and thus more legible? Thank you very much!

A major point of criticism concerns Table 1:

In a radiotherapy study, an "unknown" cannot be accepted in the definition of the radiation technique (VMAT, cyberknife, etc.). The radiotherapy documentation must clearly indicate which technique was used to treat which target volume. The same applies to hyperthermia: here, the documentation must show how often hyperthermia was performed. Otherwise, how can the authors ensure that hyperthermia took place at all in the case of "unknown"? Here I urgently request that the patient files be studied again, or - in the unlikely event of irreversible data loss - that the publication be cleansed of the affected patients, i.e. that these patients be excluded from the analyses / publication.

The authors should discuss why hyperthermia was performed AFTER radiotherapy (and not BEFORE radiotherapy), i.e. why radiation was not applied to the hyperthermic tissue.

Some further suggestions for the “discussion”:

“Although this potency is well-studied in breast cancer, showing again higher complete response rates when hyperthermia is added to radiotherapy [19].” -> delete “although”

There are few studies reporting on the addition of hyperthermia to radiotherapy in recurrent or SP. -> add “or SP HNC”

Round 2

Reviewer 1 Report

The authors answered my questions and modified the manuscript, which is greatly improved (data in the tables were corrected for example). Concerning Kaplan Meier analysis for LC and OS, I understand that the authors plan to add a control group without Hyperthermia in a future study. However, I still think this control is essential to publish in a journal of the rank of Cancers.

Author Response

We thank Reviewer 1 for the kind words. We agree with Reviewer 1 that  the lack of a control group is scientifically not so sound. In our setting, however, it was not feasible to do a randomized study, as the hyperthermia is reimbursed in the reirradiation setting. We feel however, that that RF hyperthermia in the head and neck region is hardly reported in literature and therefore the presented data are of high value to the head and neck as well as the hyperthermia scientific community. 

Reviewer 2 Report

The authors underlined in the text the criticalities of the group under consideration, which however remain quite unresolved. The need for a larger study with other types of patients is imperative. However, I believe it is a very small step forward for the treatment of patients with this collar and therefore I suggest to publish the work. Please check the entire manuscript for typos and grammar errors

Author Response

We thank Reviewer 2 for the kind words regarding our manuscript, we agree with Reviewer 2 that this a small step forward and we are working to bring our HN hyperthermia treatment to the primary setting. We have further checked the manuscript for typos and grammar errors.